# Violence on the Job: The Experiences of Nurses and Midwives with Violence from Patients and Their Friends and Relatives

**DOI:** 10.3390/healthcare8040522

**Published:** 2020-11-30

**Authors:** Jacqueline Pich, Michael Roche

**Affiliations:** School of Nursing and Midwifery, University of Technology Sydney, Sydney, NSW 2007, Australia; Michael.roche@uts.edu.au

**Keywords:** violence, aggression, workplace violence, nursing, midwifery

## Abstract

Violence in healthcare is recognised as a significant workplace issue worldwide, with nurses recognised as the profession at greatest risk. The purpose of this study was to explore nurses’ and midwives’ experiences of violence in different clinical areas, work sectors and geographical regions. A cross-sectional design was employed to survey the membership of the New South Wales Nurses and Midwives’ Association about their experiences with violence from patients and/or their friends and relatives in their workplace. A total of 3416 participants returned a completed questionnaire and more than three-quarters of had experienced an episode of violence in the preceding six months. Participants working in the public health sector reported significantly more physically violent behaviours than their colleagues in the private sector. No statistically significant difference between the rates of violence (overall) was identified between different geographical areas. Violent behaviours were reported across all clinical settings, with emergency departments, mental health and drug and alcohol settings reporting the highest proportion of episodes. The results of this large study highlight the high levels of violence that nurses and midwives experience in the workplace across all sectors of employment, geographical regions and clinical settings.

## 1. Introduction

Workplace violence, in general, is reported to be increasing in both frequency and severity and within this context the health industry has been identified as one of the most violent workplaces to work [1]. Healthcare workers are more likely to be attacked at work than prison guards and police officers [2] and both the World Health Organization and International Council of Nurses have recognised violence in healthcare as a significant global issue [3].

There is a growing body of literature focused on the topic of violence in healthcare and within this context nurses have been identified as the profession at highest risk [4]. Nurses are regularly exposed to verbal abuse and physical violence in the course of their work. These high levels of violence have resulted in a desensitisation on the part of many nurses to the point where violence has become an expected and accepted part of their job [5]. Verbal and physical abuse are regarded as occupational hazards and rationalized as “part of the job” of being a nurse. [6]

Australian data on violence in healthcare settings, and in particular hospitals, highlights a steady increasing over the past few decades. For example in the period 2015 to 2018 assaults in hospitals increased by 48% in Queensland and 44% in New South Wales (NSW). In Victoria the number of nurses assaulted in health settings increased by 60% over the same period [7]. A recent study of regional nurses in Australia reported that violence towards nurses is occurring on daily basis and increasing in frequency [8].

Patients have been identified as the most common perpetrators of this violence, and this includes the parents/carers of pediatric patients [9]. Relatives and those accompanying patients have also been identified as being responsible for episodes of violence [10] and this is particularly evident in research originating from non-western countries [11].

The occurrence of patient-related violence varies substantially between clinical environments, with the specialties of emergency, aged care and mental health reporting the highest rates of violence [12].

The impact on healthcare staff is significant and can be long-lasting. Both physical and psychological symptoms are present including those associated with post-traumatic stress disorder such as insomnia, anxiety, nightmares and flashbacks [13]. Staff also suffer physical harm and in extreme cases death. A related, growing, concern is sexual harassment and assault of healthcare staff by patents [14,15].

A number of patient-specific precipitants and antecedents for violence have been identified. These include a previous history of violence, substance abuse, alcohol intoxication, cognitive impairment, and mental health issues [9]. However, much of this work has focused on the experiences of healthcare staff and the precipitants and antecedents thereby identified are consequently based on the perceptions of staff.

### Definition of Violence

For the purposes of this study violence was defined as “refers to any incident or behaviour in which staff feel abused, are threatened or assaulted in circumstances arising out of, or in the course of, their employment including verbal, physical or psychological abuse, threats or other intimidating behaviours, intentional physical attacks, aggravated assault, threats with an offensive weapon, sexual harassment and sexual assault.” [16] (p. 3).

## 2. Materials and Methods

This study utilised a cross-sectional design to survey the membership of the NSW Nurses and Midwives’ Association (NSWNMA), the principal industry body for nurses and. An online survey was used to investigate the experiences of violence from patients and/or friends or relatives, reported by nurses and midwives. The survey was distributed broadly to Registered (licensed) Nurses, Registered Midwives, Enrolled Nurses (similar to LPN/LVN in North America) and Assistants in Nursing (unregulated healthcare workers also referred to as a nurse assistants, nurse aides, and by other titles). New South Wales is Australia’s most populous state with approximately 7.4 million residents. The median population age is 38 with 16.3% of the population over 65. Approximately two-thirds of residents (65.5%) were born in Australia, and the population reflects a wide multicultural diversity [17]. In 2015 there were 1036 nurses and midwives per 100,000 residents, most employed in metropolitan regions. Nationally, the average age of nurses and midwives is 44.4 [18].

### 2.1. Study Aims

The study aimed to explore the experiences of nurses and midwives in different clinical areas, work sectors and geographical regions. The specific objectives of the study discussed in this paper were:To describe the frequency of individual nurses and midwives’ exposure to violence;To describe the different types of violence experiences by nurses and midwives;To compare nurses and midwives’ exposure to violence across clinical areas, work sectors and geographical regions.

### 2.2. Study Instrument

Nurses and midwives were asked about their experiences of violence from patients and/or friends and relatives in their workplace using a questionnaire specifically developed for this study. In a review of the evidence the researchers were unable to identify a standardized instrument that would have addressed the aims of this project, therefore a bespoke questionnaire was developed. The questions included were derived from a review of the literature and reviewed by a panel of expert nurses prior to distribution. This questionnaire gathered information on respondent characteristics (age group, gender, employment sector, clinical specialty, region, position, and experience), and the experience of violence (type of violence, direct or witnessed).

Respondent characteristics used widely adopted classifications derived from the Australian National Health Data Dictionary and other sources.

Regions are defined as follows:Major city: geographic distance imposes minimal restriction upon accessibility to the widest range of goods, services and opportunities for social interaction;Inner and outer regional: areas where geographic distance imposes some/moderate restriction upon accessibility to the widest range of goods, services and opportunities for social interaction;Remote: and very remote areas where geographic distance imposes a high restriction upon accessibility to the widest range of goods, services and opportunities [19].

Employment sector was defined as follows:Public: employment by the NSW Ministry of Health;Private: employment by a private/for profit organization;Not for profit: employment by a not-for-profit organisation.

Experience of violence was a binary response (yes/no). It specifically requested information about violence from patients and their friends or relatives in both the preceding week and six-months since completion of the questionnaire, worded as “Have you been involved in one or more episodes of violence in the last week?” and “Have you been involved in one or more episodes of violence in the last six months?”. Participants were also asked to indicate the type of violence: verbal abuse and/or non-physical behaviours and/or physical abuse/violence. They were then asked to “indicate the following types over verbal abuse and/or non-physical behaviours you have observed and/or witnessed during episodes of violence” and provided a dropdown list of behaviours to choose from, from which they were instructed to select all that applied. This included an option for “other” where participants were asked to elaborate in an open-ended response. The same question was asked in relation to physical behaviours observed and/or witnessed. The list of behaviours provided was developed from a search of the literature on the topic.

### 2.3. Ethical Considerations

Questionnaires were anonymous with no individually identifying material collected. Participants were provided with detailed information about the intended use of the findings and completion of the online questionnaire implied informed voluntary consent. Ethical approval for the study was granted by the Human Research Ethics Committee of the University of Technology, Sydney (ETH20-5169).

### 2.4. Data Analysis

The statistical software program SAS v9.4 (SAS Institute, Cary, NC, USA) was used to analyse the study data, with the support of a statistician. Descriptive data analysis was used to describe the frequencies of answers to non-open-ended questions. Groups of interest were identified based on responses to the questionnaire and were compared using Chi-square and Fisher’s exact test, the latter being used when 20% or more of the expected frequencies were less than five. Statistical significance was set at 0.05, but was not calculated for variables with excessive low counts.

## 3. Results

The membership of the New South Wales Nurses and Midwives’ Association was 62,954 at the time of sampling. A total of 3416 completed questionnaires were returned by eligible participants, a response rate of 5.5%. There was an incomplete response to some survey questions and, consequently, the denominator varies in the results.

### 3.1. Participant Summary

The majority of respondents were female (87%) registered nurses (77%) who were aged between 26 and 65 years (89%). Almost 50% of respondents worked in a metropolitan area (46%). Most nurses who did not work in a major city worked in outer and inner regions of NSW (25%, respectively). A majority of nurses worked in the public sector (78%), followed by the private sector (16%), and not for profit (7%) (Table 1).

Participants were drawn from a variety of clinical areas, primarily medical-surgical (*n* = 750, 24%), mental health (*n* = 547, 18%), aged care (*n* = 483, 16%) and emergency (*n* = 297, 10%). Thirteen percent (*n* = 401) of participants visited patients/clients in their homes as part of their work, and 7% (*n* = 224) worked in midwifery.

### 3.2. Incidence of Violence

Of the over 3000 participants who responded to questions regarding violence, nearly half (*n* = 1454, 47%) had experienced an episode of violence in the week prior to completing the survey. Most (*n* = 2460, 79.3%) had experienced violence in the previous six months. Where violence had been encountered, there was a wide range in the number of episodes, from 1 to 100, although most (*n* = 2014, 81%) reported between 1 and 20. Sixty-three (2.5%) had experienced more than 80 episodes of violence (Figure 1).

One quarter (*n* = 606, 24%) of participants reported physical abuse/violence. More than half reported being grabbed (60%); hit (60%); spat at (53%); kicked (53%); pushed (53%); or punched (52%) (Table 2), while 55% reported destructive behaviour such as hitting a table or other object. In addition, 805 participants reported inappropriate physical or sexual contact and 35 indicated that they had been sexually assaulted. Participants provided additional detail in comments, including:“…Liquid chlorpromazine thrown in my face…”“…grabbing and shaken and slapped on face…”“…stalking…:“…Intentionally coughing in my face accompanied by statements such as “I hope you get this”…”“…Grabbed by the waist and pinned to the bed rail by male patient…”“…Throwing any object that is available to hit staff…”“…lighting a fire in the department…”.

Verbal or non-physical violence was also common, with 1888 (76%) of participants experiencing an episode in the previous six months. The most common types were swearing; rudeness; anger; shouting; and making unreasonable demands, while a quarter had experienced sexually inappropriate behaviour. The use of social media (5%) and taking of photographs (9%) was reported by 14% of participants. (Table 2). Participants provided additional details in comments, including:“…ganging up behaviour; enticing others in ganging up against nursing staff…”“…filming…”“…Just released ex prisoner (Manslaughter) allowed to harass staff without boundary by Mental Health Administration…”“…Demeaning inappropriate personal questions…”“…death threats and threats to children and colleagues…”“…racism…”“…Video on phones of family members…”“…threatened with patient’s blood…”“…told…if his terminally ill mother died he would kill us all, she did AND he kept coming back to the hospital and threatening us forcing us to lock ourselves in rooms…”.

Although there was no statistically significant difference between the rates of violence overall between different geographical areas (Table 3), some physically violent behaviours were more common in regional locations relative to metropolitan or remote areas. These included being grabbed (39% vs. 32% and 25%, *p* < 0.01); being hit (35% vs. 31% and 25%, *p* = 0.04); having a body part twisted and pulled (24% vs. 18% and 13%, *p* < 0.01). Conversely, some behaviours were more common in metropolitan areas, specifically the use of a traditional weapons such as a knife (11% vs. 4.6% and 4.8%, *p* = 0.02); the use of non-traditional weapons such as ‘sharps’ (77% vs. 70% and 71%, *p* = 0.03) and choking or strangling (15% vs. 7.7% and 4.8%, *p* = 0.02).

Participants employed in the public sector reported more physically violent behaviours than their colleagues in the private sector, in a result that was statistically significant. More than half of the participants from each clinical area reported experiencing violence, with those working in the emergency department, mental health and drug and alcohol settings reporting the highest proportion of episodes of violence (Table 3).

Males reported significantly more instances of violence and midwives fewer (Table 4). Highly experienced nurses and midwives reported fewer episodes relative to their less experienced colleagues, which was consistent with age group findings.

## 4. Discussion

This aim of this study was to survey the members of the NSWNMA on their experiences with episodes of violence from patients and or their relatives and friends in the workplace. This paper reports one of the largest surveys on this topic in this population, with a sample size of 3416, and this high response rate demonstrates the significance of this issue to NSW nurses and midwives. The respondent profile in this study was consistent with the Australian and NSW nursing workforce in terms of age, gender, and position [18]. The majority of participants were split between metropolitan (*n* = 1487) and regional (*n* = 1606) locations with the remaining 161 participants situated in remote areas of the state.

### 4.1. Incidence of Violence

The majority of participants had experienced some form of violence during the week and six months prior to data collection. Nurses and midwives reported between one and 100 episodes in the previous six months, with most indicating that they had experienced between one and 10 episodes and 2% reporting more than 10 episodes.

Participants had experienced episodes of both verbal and physical violence. Non-physical or verbal violence was the most common type, reported by more than three quarters of participants. These results are consistent with previous studies where rates of up to 90% have been reported [20]. High levels of verbal abuse against nurses and midwives are a significant global issue, and have been reported in a number of studies from around the world [21,22,23,24].

A large study of emergency department nurses (*n* = 7169) found that that 43% had experienced verbal abuse, during a one-week period [25]. Similarly, 38% of American nurses in a large study (*n* = 6300) had reported at least one episode of threat, sexual harassment or verbal abuse [26]. An Australian survey of Australian emergency department nurses (*n* = 537) reported that 87% had experienced violence in the last six months and 40% in the previous week [27]

It is important to acknowledge that while these studies report on data collected over a range of time periods, all of them describe unacceptably high rates of violence.

### 4.2. Physical Violence

Physical violence had been experienced by a quarter of those surveyed. The most common behaviours ranged from grabbing to hitting and punching, and included spitting, kicking, pushing, and other destructive behaviour. Nurses and midwives employed in the public sector reported a higher rate of these behaviours than those working in the private or not-for-profit sectors. This is consistent with other studies that reported nurses being grabbed, pushed or shoved, punched, head-butted, slapped or otherwise hit, kicked or scratched, spat on, and having hair pulled [25].

More than 800 of those surveyed reported inappropriate physical or sexual contact, with 35 respondents indicated that they had been sexually assaulted. Sexual harassment against nurses and midwives has emerged as a significant issue [28]. A meta-analysis of the worldwide prevalence of sexual harassment towards nurses (*n* = 52,345 participants) reported that the prevalence of sexual harassment towards nurses during their nursing career was 53% and 13% in the preceding 12 months [29].

The comments provided in the open-ended responses by participants highlighted serious violent behaviours with the potential to cause serious injury and even loss of life, for example the use of hazardous substances such as liquid chlorpromazine, blood and the opportunistic use of hospital equipment as weapons.

### 4.3. Verbal Violence

The most common verbal behaviours reported were swearing, rudeness, anger, shouting and making unreasonable demands and these were experienced at higher rates by participants employed in the public sector compared to the private and not-for-profit sectors. These findings are consistent with previous studies on the topic where swearing or “being cursed at” is the most common type of verbal abuse [25,30,31]. The use of abusive social media and non-consenting photographs was a significant issue for metropolitan nurses and midwives and demonstrates how pervasive aggressive behaviours are, extending beyond the confines of the work environment.

Swearing has been identified as the most common form of verbal abuse [32] and demeaning swearing has been identified as the most offensive form of verbal aggression, particularly for female nurses [33]. This included gendered and sexualised insults, judgements, threats or suggestions and demeaning statements often made in public spaces in front of others in an attempt to draw negative attention to those nurses’ targeted [10].

### 4.4. Violence and Participant Characterics

A downward trend was noted for years of experience and episodes of violence with less experienced nurses and midwives more likely to have experienced violence on the job. This trend was repeated with age with 94% of participants aged 18–25 years reporting an episode of violence in the preceding six months as opposed to 68% in those aged over 65 in a finding that was significant. These findings are consistent with the literature on the topic which consistently reported that younger, less experienced nursing staff were at higher risk of verbal and physical violence [25].

Nursing and midwifery are female dominated professions with women comprising approximately 86% of the nursing workforce and 99.06% of the midwifery workforce in Australia [34]. Participants of both genders reported high levels of violence with males more likely to experience an episode of violence than females. This finding was for any type of violence, and was not broken down to verbal and physical violence. The body of evidence on the topic indicates that female nurses are more likely to experience verbal abuse and non-physical violence than their male colleagues, with male nurses reporting higher levels of physical violence. The authors of an Australian study discussed the use of male nurses as “de facto bodyguards for their female colleagues” and theorized that this exposed them to higher levels of physical violence [8]. In a systematic review of exposure to physical violence male nurses were reported to experience physical violence at a higher rate than female nurses [35].

### 4.5. Violence and Clinical Area and Work Sector

More than half of the participants surveyed across all clinical areas had experienced an episode of violence in the preceding six months. These numbers were highest for the specialties of emergency, drug and alcohol and mental health in a finding that is consistent with the literature on the topic [9]. However participants working in clinical areas traditionally viewed as low-risk areas for violence such as midwifery, community health and family and child health also reported high levels of violence.

Participants employed in the public sector were more likely to have been involved in an episode of violence in the preceding six months compared to those employed in the private and not-for-profit sectors. No differences were identified between the public and private sectors for episodes of verbal abuse, however participants employed in the public sector were found to have experienced physical violence at higher levels. In remote areas no difference was identified between participants employed in the public and private sector.

Verbal and physical violence were both found to be more common for participants employed in the private sector compared to the public sector in inner-regional areas. No differences were noted for physical violence between these employment sectors in outer-regional areas, however, verbal abuse was experienced at higher levels by those employed in the public sector.

### 4.6. Limitations

The authors acknowledge the potential for non-response bias and self-selection bias due to the type of data collection. A retrospective approach was employed, and participants asked to self-report on their experiences for a period of six months prior to completing the questionnaire. This approach may lead to an increased risk of recall bias although the authors believe this is similar to many similar studies, including those to which findings have been compared. A related issue is under-reporting, which although also consistent with similar studies, is of greater concern. However it is reasonable to expect that responses here may be more representative than reporting based on routine monitoring and voluntary reporting of violent episodes in healthcare organisations, where organizational cultural issues may be at play. Nonetheless, findings here should be considered with this issue in mind. The authors acknowledge the low response rate of 5.5%, however, the large sample size representative of the target population mitigates this.

## 5. Conclusions

The results of this large study highlight that nurses and midwives experience workplace violence from patients and/or their relatives and friends at unacceptably high levels and are subjected to verbal and physical violence on a regular basis. They are exposed to a range of violent behaviours, including sexual harassment and assault, which place them at significant risk of physical and psychological injury. This is compounded by the fact that some types of violence extend beyond the workplace through the use of social media and threatening behaviours. These findings were consistent across all clinical areas, work sectors and geographic regions. The emergency department and mental health settings have traditionally been identified as the highest-risk areas for violence, however, this study found that nurses and midwives were at risk of violence from patients and/or their relatives and friends regardless of work setting, and this extended to those working in the community. Therefore, violence prevention and management strategies should encompass all workplaces that nurses and midwives are engaged in.

The public health sector was found to have the greatest risk of potential violence compared to the private and not-for-profit sectors, however more than half of nurses and midwives working in all employment sectors reported exposure to violence. Similarly the majority of nurses working across all geographical regions reported violence at unacceptably high levels.

The findings of this study serve to illustrate that all nurses and midwives are at risk of experiencing violence in their workplace from patients and/or their relatives and friends.

### Future Directions for Research

Research on the experiences of nurses and midwives with workplace violence from patients and their relatives and friends has largely been focused on clinical areas perceived to be at higher risk, for example the emergency department, aged care and mental health settings. This study identified all clinical environments as at-risk for potential violence, therefore future studies should be expanded to include these were possible.

The experiences of the general public who are directly or indirectly involved with episodes of violence in healthcare settings is largely absent from the existing body of research and may enable researchers to better understand the phenomena and inform future violence prevention training of health professionals.

## Figures and Tables

**Figure 1 healthcare-08-00522-f001:**
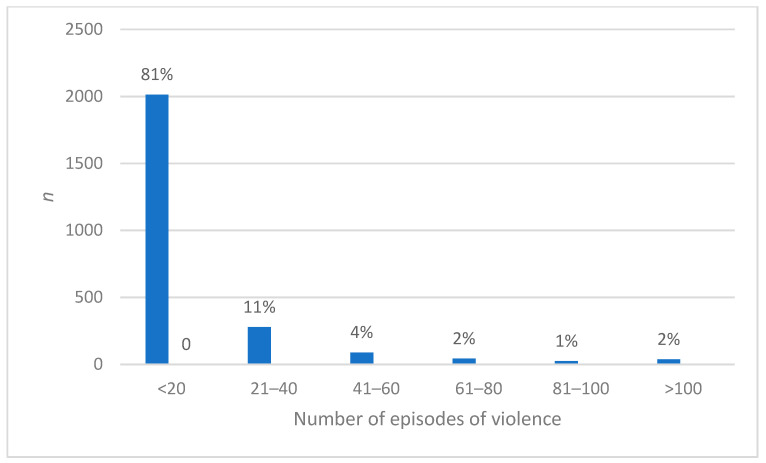
Episodes of violence experienced by nurses and midwives.

**Table 1 healthcare-08-00522-t001:** Participant characteristics.

Variable	Category	Total (%)
Employment sector (*n* = 3327)	Public	2585 (78%)
	Private	517 (16%)
	Not for profit	225 (7%)
Region (*n* = 3254)	Major city (metropolitan)	1487 (46%)
	Outer regional	804 (25%)
	Inner regional	802 (25%)
	Remote	140 (4%)
	Very remote	21 (1%)
Gender (*n* = 3339)	Female	2909 (87%)
	Male	409 (12%)
	Other	21 (1%)
Position (*n* = 3357)	Registered Nurse	2598 (77%)
	Enrolled Nurse	377 (11%)
	Registered Midwife	224 (7%)
	Assistant in Nursing	158 (5%)
Age group (*n* = 3347)	18–25 years	235 (7%)
	26–35 years	531 (16%)
	36–45 years	583 (17%)
	46–55 years	901 (27%)
	56–65 years	981 (29%)
	>65 years	116 (3%)

NB: Different missing data per variable.

**Table 2 healthcare-08-00522-t002:** Types of violence.

Physical Violence (*n* = 1957 *)	*n* (%)	Verbal/Non-Physical Violence (*n* = 2761 *)	*n* (%)
Grabbing	1179 (60%)	Swearing	2310 (84%)
Hitting	1166 (60%)	Rudeness	2214 (80%)
Destructive behaviour e.g., punching safety glass, table etc.	1084 (55%)	Anger	2181 (79%)
Spitting	1038 (53%)	Shouting	2046 (74%)
Kicking	1032 (53%)	Making unreasonable demands	2008 (73%)
Pushing	1029 (53%)	Insulting/questioning professional ability e.g., incompetent, incapable, threatening registration	1767 (64%)
Punching	1011 (52%)	Sarcasm	1674 (61%)
Scratching	809 (41%)	Name calling	1545 (56%)
Grabbing and twisting a body part	719 (37%)	Stepping into personal space	1249 (45%)
Throwing/struck with an object	664 (34%)	Threatening comments—to self, family or property	1109 (42%)
Use of non-traditional weapons e.g., sharps, IV poles, chair	583 (30%)	Ridicule in front of others	1109 (40%)
Biting	575 (29%)	Unjustified criticism	1064 (39%)
Inappropriate physical contact	558 (29%)	Symbolic violence e.g., punching/hitting glass/desk at triage	1032 (37%)
Body fluids thrown e.g., blood, urine, faeces	388 (20%)	Staring	899 (33%)
Pulling hair/jewellery/clothing	377 (19%)	Gesturing	890 (32%)
Restraining/immobilising staff	250 (13%)	Sexually inappropriate behaviour	699 (25%)
Inappropriate sexual conduct	247 (13%)	Berating	685 (25%)
Damage to personal property e.g., tyres slashed	175 (9%)	Formal complaints without cause	512 (19%)
Choking/strangling	168 (9%)	Rumour mongering	462 (17%)
Use of a traditional weapon e.g., knife	105 (9%)	Taunting	461 (17%)
Sexual assault	35 (2%)	Taking photographs	259 (9%)
		Use of social media	128 (5%)

* multiple responses per variable allowed.

**Table 3 healthcare-08-00522-t003:** Episodes of violence, by work sector and clinical area.

Variable	No (*n* = 632 *)	Yes (*n* = 2460 *)	X^2^	*p*
Employment Sector				
Public	436 (18%)	1973 (82%)	41.3	<0.01
Private	146 (31%)	323 (69%)		
Not for Profit	39 (21%)	151 (79%)		
Clinical Area				
Midwifery	87 (42%)	121 (58%)	268.5	<0.01
Medical/Surgical	129 (18%)	561 (82%)		
Emergency Department	18 (6%)	264 (94%)		
ICU, HDU or CCU	28 (15%)	159 (85%)		
Aged care	89 (21%)	335 (79%)		
Drug and Alcohol	3 (6%)	44 (94%)		
Mental health	33 (6%)	481 (94%)		
Community health	52 (41%)	74 (59%)		
Family/Child Health	47 (44%)	60 (56%)		
Perioperative	59 (36%)	106 (64%)		
Rehabilitation/disability	18 (20%)	73 (80%)		
Region				
Major city	295 (21%)	1081 (79%)	8.9	0.06
Inner Regional	135 (18%)	599 (82%)		
Outer Regional	137 (18%)	605 (82%)		
Remote	26 (19%)	111 (81%)		
Very Remote	7 (41%)	10 (59%)		

Note: * multiple responses per variable allowed; ICU (Intensive Care Unit); HDU (High Dependency Unit); CCU (Cardiac Care Unit).

**Table 4 healthcare-08-00522-t004:** Episodes of violence and participant characteristics.

Variable	No (*n* = 632)	Yes (*n* = 2460)	X^2^	*p*
Experience				
1–5 years	70 (12%)	526 (88%)	62.9	<0.01
6–10 years	68 (16%)	351 (84%)		
11–20 years	122 (20%)	491 (80%)		
21–30 years	114 (21%)	422 (79%)		
>31 years	257 (28%)	670 (72%)		
Age Group				
18–25 years	13 (6%)	203 (94%)	61.7	<0.01
26–35 years	71 (15%)	417 (85%)		
36–45 years	99 (18%)	440 (82%)		
46–55 years	189 (23%)	648 (77%)		
56–65 years	223 (25%)	676 (75%)		
>65 years	34 (32%)	71 (68%)		
Gender				
Female	576 (22%)	2096 (78%)	19.3	<0.01
Male	46 (12%)	339 (88%)		
Other	2 (25%)	6 (75%)		
Position				
Registered Nurse	447 (19%)	1941 (81%)	50.5	<0.01
Registered Midwife	82 (39%)	126 (61%)		
Enrolled Nurse	73 (21%)	282 (79%)		
Assistant in Nursing	30 (21%)	111 (79%)		

NB: Different missing data per variable.

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
