# Peer review of "Violence on the Job: The Experiences of Nurses and Midwives with Violence from Patients and Their Friends and Relatives"

_healthcare, 2020, doi:10.3390/healthcare8040522_

Round 1
Reviewer 1 Report
Dear authors.
Thank you very much for your research.
I think that there are some aspects that must be considered for publishing.
First of all the theoretical frame is too brief. I have missed some information about why are nurses in danger, and why violence appears over this group. There are some theories that could explain this situation.
Besides, your method section should be improved. You should use a validated instrument, in order to obtain reliable measures. If you are assessing types of violence, I am sure that you could find an instrument that fits.
If you want to create a taxonomy of violence in workplace, I am sure you could find an instrument and theoretical frame that support it.
Conclusion section is very brief as well.
Thank you very much.
Reviewer 2 Report
XXXX
Violence on the job: The experiences of nurses and midwives with violence from patients and their friends and relatives.
Healthcare-970118 October 30, 2020
The authors present an interesting, timely, important and well-written study on violence against nurses, mid-wives. While the response rate is low (as noted) it is an important view of what is happening to nurse and mid-wives in regard to violence on the job.
Was race noted and just left off the demographic table?
I have very minor changes to the manuscript and they appear below.
In the abstract remove the use of “of” line 16 and 18.
Line 111 free-text answers might better be stated as “open-ended responses.”
Lines 159-174 include statement made by participants. However, these responses to seem to be lumped into one paragraph. Is there a different way to format these responses, therefore making it more decipherable for the reader? E.g.,
Participants provided additional detail in comments, including
Liquid chlorpromazine thrown in my face
…grabbing and shaken and slapped on face
Additionally, I was not sure what the numbers/letters mean after the statement. Is that the participant identification? If so, please remove (not necessary). If it has some other meaning please explain.
The other tables are excellent.
Line 246 Should read: such as liquid chlorpromazine, and blood, and the opportunistic use…
Line 256 has an extra period
Line 271 Similarly no differences were (add the s)
Since 87% of your participants were female, I think I little bit more discussion of gender-based violence and societal tolerance of this crime is warranted in your conclusion.
XXXXXXXXXXXXXXXXXXXXXXXXXXXXXXXXXXXXXXXXXXXXXXXXCC
Reviewer 3 Report
Provide example items from the survey.
Please add an about the researcher section so you tell us more about you as the researchers and your connection to this study. How does this align with personal interests, professional work, etc., to help the reader place you directly in the center of your work?
Please add future directions for this research.
Round 2
Reviewer 1 Report
Dear authors.
I have reviewed your changes.
I think that your paper have gained in readibility and you have addressed the sugestions I did to you.
I hope that you found them helpful.
Good luck in publishing.